# A Nomogram Model for Predicting the Polyphenol Content of Pu-Erh Tea

**DOI:** 10.3390/foods12112128

**Published:** 2023-05-25

**Authors:** Shihao Zhang, Chunhua Yang, Yubo Sheng, Xiaohui Liu, Wenxia Yuan, Xiujuan Deng, Xinghui Li, Wei Huang, Yinsong Zhang, Lei Li, Yuan Lv, Yuefei Wang, Baijuan Wang

**Affiliations:** 1College of Mechanical and Electrical Engineering, Yunnan Agricultural University, Kunming 650201, China; 18637905872@163.com; 2Yunnan Organic Tea Industry Intelligent Engineering Research Center, Yunnan Agricultural University, Kunming 650201, China; yang_960204@163.com; 3College of Tea Science, Yunnan Agricultural University, Kunming 650201, China; lxh2006i@163.com (X.L.); yuanwenxia2023@163.com (W.Y.); 2021066@ynau.edu.cn (X.D.); huangwtea@ynau.edu.cn (W.H.); lilei53454@163.com (L.L.); 4China Tea (Yunnan) Co., Ltd., Kunming 650201, China; shengyubo@cofoco.com; 5International Institute of Tea Industry Innovation for “the Belt and Road”, Nanjing Agricultural University, Nanjing 210095, China; lxh@njau.edu.cn; 6College of Foreign Languages, Yunnan Agricultural University, Kunming 650201, China; zys30507521@163.com (Y.Z.); luke_1874@163.com (Y.L.); 7College of Agronomy and Biotechnology, Zhejiang University, Hangzhou 310013, China

**Keywords:** abiotic stress, tea polyphenol, LASSO regression, nomogram model, visualized, prediction system

## Abstract

To investigate different contents of pu-erh tea polyphenol affected by abiotic stress, this research determined the contents of tea polyphenol in teas produced by Yuecheng, a Xishuangbanna-based tea producer in Yunnan Province. The study drew a preliminary conclusion that eight factors, namely, altitude, nickel, available cadmium, organic matter, N, P, K, and alkaline hydrolysis nitrogen, had a considerable influence on tea polyphenol content with a combined analysis of specific altitudes and soil composition. The nomogram model constructed with three variables, altitude, organic matter, and P, screened by LASSO regression showed that the AUC of the training group and the validation group were respectively 0.839 and 0.750, and calibration curves were consistent. A visualized prediction system for the content of pu-erh tea polyphenol based on the nomogram model was developed and its accuracy rate, supported by measured data, reached 80.95%. This research explored the change of tea polyphenol content under abiotic stress, laying a solid foundation for further predictions for and studies on the quality of pu-erh tea and providing some theoretical scientific basis.

## 1. Introduction

Soil is the growth substrate for tea trees [*Camellia sinensis* (L.) O. Kuntze], and so soil fertility, as the comprehensive reflection of soil indicators, determines the growth of tea trees. Moreover, soil nutrients, vital to soil fertility, weigh heavily in the quality and production of tea. It has been found that pu-erh tea has many health benefits, such as antibacterial, antidiabetic, and blood lipid-lowering activity, it has gained increasing attention from people. Sun-dried mao cha is the material for making pu-erh tea, and the main components exuding tea flavors are tea polyphenols, catechins, caffeine, and free amino acids. Tea polyphenol, a general term for polyphenols in tea, tastes bitter, is one of the main sources of the taste of tea soup [1], and can help protect against cardiovascular diseases and cancers. It was revealed in relevant studies that tea polyphenol plays a critical role against COVID-19 during the initial stage of infection [2]. Tea polyphenol is a kind of polyphenolic compound. It is a natural antioxidant, preservative, and food additive and is characterized by its green, healthy, and safe qualities. Many factors influence the tea polyphenol content of fresh tea leaves, such as the temperature, light, water, and air, affecting the growth and economic production of tea trees and the formation, metabolism, and transformation of biochemical components in tea. Across different environmental conditions, the metabolites of tea plants vary considerably. Different light intensities and temperatures caused by altitudes and season changes can create variable tea polyphenol content.

Many scholars have started calculating the physicochemical components of products according to the environment and soil composition [3]. However, there is no stable and reliable prediction tool for tea polyphenol content worldwide. In the realm of tea science, Dongchao Xie and other experts made simple projections for the storage of white tea according to the change of the white tea component [4], but the accuracy is low. In the study “Prediction of Tea Polyphenols, Free Amino Acids and Caffeine Content in Tea Leaves during Wilting and Fermentation Using Hyperspectral Imaging”, Yilin Mao et al. achieved a prediction accuracy of 91% for tea polyphenol content. However, this method can only predict the tea polyphenol content in fermented tea leaves through hyperspectral imaging and cannot predict it in advance [5].

A nomogram is a visual mathematical model which combines many elements to forecast a specific result [6]. Compared to those of other forecasting models, the results of nomogram models are usually presented in a more intuitive chart form. Furthermore, their construction process is relatively rapid and does not require a large number of training data sets, thereby making them very advantageous in terms of model establishment. Jiaojiao Xie and others built a nomogram model based on only 104 sample data set, achieving an accuracy of 96.2% in predicting the clinical outcome improvement rate of COVID-19 patients [7]. Hao Chen and colleagues also used a nomogram model to analyze data from 61 patients with small cell carcinoma of the esophagus (SCCE), ultimately achieving a prediction accuracy of over 70% for patient survival [8]. With its quality of convenience and reliability, the nomogram has been extensively applied in the medical field, especially in the present intelligent diagnosis [9]. By contrast, it has been rarely used in agricultural science [10]. This study selects the same variety of tea from the same region, and constructs a multiple regression model to explore the relationship between soil quality and altitude with tea polyphenols, based on the contributions of various influencing factors (such as altitude, soil pH, organic matter and nutrient content) in the model. Meanwhile, this study has also established a prediction system. To create the prediction system for pu-erh tea polyphenol content under abiotic stress with the help of a nomogram, methods related to statistics were introduced. The system developed in this study overcomes the limitation of the nomogram model by providing two comparison methods, chart comparison and digital comparison. Chart comparison is enabled via the graphical summary module and allows users to view and compare multiple results at the same time. Digital comparison, enabled via the numerical summary module, allows users to compare the input parameters and results in greater detail and to view the numerical value of each parameter.

## 2. Materials and Methods

### 2.1. The General Situation of the Research Location

The research location of this study was Xishuangbanna, Yunnan, China—the birthplace of big leaf tea and the hometown of pu-erh tea. As this region has a warm, humid climate all year, the seasons are not divided into four; the dry season spans from November to April, while the wet season is from May to October, making it an ideal environment for the planting of a variety of teas.

The climates in Xishuangbanna are mainly a mix of the humid south subtropical climate and the north tropical climate. The tea plantations are mostly distributed in mountainous or hilly zones with lukewarm weather or humid and hot conditions. The sunshine duration of these places, which are foggy at dawn and night, is long; the altitude is 500 to 2000 m; and the annual average temperature is between 12 °C and 23 °C. In these areas, the relative humidity stays around 85%, and the annual rainfall is over 1000 mm, with the maximum being 2000-odd millimeters. The deep but loose soil here is slightly acidic (red soil, yellow soil, laterite soil) with great drainage and permeability and a pH of 4 to 6.

### 2.2. Experimental Materials

Tea leaves and soil data in this experiment were collected from the Yuecheng base (Figure 1). The tea variety used was the Mengku large leaf species, with all the trees aged over 20 years. Fresh leaves were picked under the one-bud-with-two-leaves standard and processed under the guidance of the conventional process of sun-dried mao cha [11]. In the plantation, one-bud-with-two-leaves samples were chosen from sunny and shady slopes where the top, middle, and bottom were the sampling areas each sampled three times, and all buds were kept in numbered self-sealing bags. Relevant physicochemical components were tested through the collaboration of the tea college affiliated with Yunnan Agricultural University and Yunnan Organic Tea Industry Intelligent Engineering Research Center. The moisture content was measured using a rapid method, constant weight method of aqueous extract, the gross amount of amino acids ninhydrin colorimetry, tea polyphenol content Folin phenol colorimetry, and catechin and caffeine contents high performance liquid chromatography (HPLC) established in the laboratory [12]. The determination of copper, zinc, chromium, nickel, and lead content was carried out using flame atomic absorption spectrophotometry; the determination of total phosphorus using the alkali fusion-Mo-Sb antispectrophotometric method; the determination of effective phosphorus using the sodium hydrogen carbonate solution-Mo-Sb ant spectrophotometric method; the determination of effective potassium using the combined leaching–colorimetric method; the determination of total nitrogen using the Kjeldahl method; the determination of fluoride using the ion-selective electrode method; the determination of cation exchange capacity (CEC) using the hexamminecobalt trichloride solution–spectrophotometric method; the determination of soil pH using potentiometry; the determination of arsenic and mercury using the atomic fluorescence spectrometry; and the determination of organic matter using the oxidation of potassium dichromate oxidation spectrophotometric determination of organic carbon multiplied by the constant 1.724. Each soil and tea sample was tested for composition three times, and the parameters were determined using the average and standard error of the three tests.

The area where soil samples were collected corresponded to the area where tea samples were collected and by the method that the cultivated soil within the limitation of 20 cm vertically was collected after the removal of 4–5 cm topsoil, and the altitude was documented. The soil samples were detected, and a total of 22 test substances were involved, including arsenic, chrome, lead, nickel, mercury, available cadmium, available chromium, available nickel, PH, Zn, Cu, organic matter, N, P, K, available potassium, available phosphorus, alkaline hydrolysis nitrogen, Mg, fluoride, cation, and tea polyphenol.

### 2.3. Statistical Analysis

The model was built on a computer produced by Lenovo (Beijing) Co., Ltd. in China, with an AMD Ryzen R7-5800H processor with 16 GB RAM, 1 TB solid-state drive, and a NVIDIA GeForce RTX 3060 Laptop GPU. The operating system used was Windows 10, with CUDA 11.0 version and R language version 4.1.2. In total, 22 influential factors were taken into consideration, including altitude, arsenic, chrome, lead, nickel, mercury, available cadmium, and available chromium, among others. Among these factors, eight, namely, altitude, nickel, available cadmium, organic matter, N, P, K, and alkaline hydrolysis nitrogen were initially defined as powerful influences on tea polyphenol content through logistic regression [13]. The standard of *p* < 0.05 indicated that the figures were statistically significant [14].

The LASSO (least absolute shrinkage and selection operator) regression method is a biased estimation addressing multicollinearity data to realize variable selection. It retains the advantages of subset shrinkage by reducing the variable set and constructing a penalty function to compress the coefficients of the variables and make some regression coefficients become 0 [15]. During the sifted stage of the LASSO regression model, 3 easily accessible variables, altitude, organic matter, and P, were selected to construct the nomogram model.

To test the accuracy and stability of the model, ROC curves and calibration curves were plotted in this study [16]. The performance of the nomogram model could was by calculating the AUC (area under curve) of the ROC curve, and the calibration curve reflected the consistency between the actual and predicted results. Statistically, if the area is over 0.75, the nomogram model has a sound predictive ability, and if the area is 0.5–0.75, it is still acceptable. To facilitate the application for future users, a visualized prediction system for pu-erh tea polyphenol content was created based on the generated nomogram model [17].

## 3. Results

### 3.1. Single-Factor Analysis

According to the 22 factors, such as altitude, arsenic, chrome, lead, nickel, mercury, available cadmium, and available chromium, and their corresponding tea polyphenol contents, the data set was randomly divided into the training set and validation set (there were 84 sets of data in this dataset). In the training set, logistic regression was applied to compare and analyze tea polyphenol contents and the 22 variables [18] (Table 1). The results revealed that altitude, nickel in soil, available cadmium, organic matter, N, P, K, and alkaline hydrolysis nitrogen had significant effects on tea polyphenol contents (*p* < 0.05).

Logistic regression analysis is a generalized linear regression analysis model whose derivation and calculation are similar to the process of regression. In practice, it is mainly applied to address binary classification or multiclass classification problem. The model is trained with given *n* sets of data (training set), and one or multiple sets of data (test set) are classified after the training. The main idea is to establish a relevant regression formula for the classification boundary according to the existing data and classify it in order. It is a posterior probability distribution, which, from the perspective of probability, has better robustness to abnormal data than does the perceptron model. Logistic regression, compared with general linear regression, has a sigmoid function, an S-type function, which is generally reckoned to be a threshold function in the neural network. The equation of sigmoid and its derivative to *x* are as follows.
(1)$$S\left(x\right)=\frac{1}{1+e^{-x^{’}}}$$
(2)$$S^{’}\left(x\right)=\frac{e^{-x}}{{\left(1+e^{-x}\right)}^{2}}=S\left(x\right)(1-S\left(x\right)),$$

### 3.2. Model Construction Factor Selection

This study screened the necessary factor amounts and types for the construction of the prediction model by building the LASSO model. Among 22 variables, 3 strongly related ones, namely altitude, organic matter, and P, were selected based on the training cohort of LASSO regression (Figure 2). From the coefficient distribution chart of LASSO regression, the ordinate (variable coefficient) is continuously compressed under the influence of the penalty term with the increase of the abscissa (penalty coefficient) until it is compressed to 0. During cross-validation, bootstrapping was used in this study, with two dashed lines representing the two optimal values of *λ*. The left one expressed the minimum mean square error of the longitudinal axis, and the right one the double standard error of the minimum mean square error. The more complicated the model is, the more overfitting it tends to produce. Therefore, the value of *λ* at double standard error was used in this study.

The LASSO regression model not only sifts variables but also effectively optimizes the complexity to some extent to reduce the types of dependent variables required in the prediction model. Screening variables is aimed at introducing the selected ones into the nomogram prediction model to optimize performance parameters. The adjustment for the complexity of the model was achieved using parameter control to avoid overfitting. For the construction of the nomogram model, the more variables there are, the more intricate the model and the more accurate the result would be to a certain extent. However, more overfitting would be produced during the process. The loss function of LASSO regression is calculated as follows:(3)$$J\left(\theta \right)=\frac{1}{2m}\sum_{i=1}^{m}{\left(h_{\theta }\left(x^{\left(i\right)}\right)-y^{\left(i\right)}\right)}^{2}+\lambda \sum_{j=1}^{n}|\theta_{j}|,$$

In this function, *λ* is the regularized parameter, which determines the complexity of the LASSO model. The higher the value of *λ* is, the fewer variables would be sifted.

The principle of bootstrapping is to choose some samples from the given data sets and repeat the experiment to obtain many different data sets to ultimately replace the overall distribution with the empirical distribution of the samples. Through random sampling with replacement to obtain a sufficient number of samples from original data sets, the statistics to be estimated were calculated based on selected samples. After 2000 times of repetition, variance and distribution were computed according to the calculation results.

### 3.3. The Construction of the Nomogram

The multivariate analysis results of logistic regression produced using RStudio (4.1.2) software are listed in Table 2.

On the basis of multifactor regression analysis, the nomogram model integrated multiple dependent variables and was drawn on the same plane in proportion with line segments with scale lines to reveal the relationship between the variables in the model. It pictured a multifactor regression model according to the value of the regression coefficient in the model (the degree of influence of each dependent variable on the change of the outcome variable), and each value of each dependent variable was assigned. Lastly, the predicted value of the outcome variable was acquired through the function of the total score and the outcome variable.

The nomogram model was drawn in RStudio based on the coefficients from the analysis (Figure 3), in which the unit of organic matter was g/kg, and that of P was mg/kg. During the prediction of the model, a total score sourced from the sum of each score of altitude, organic matter, and P corresponding to the respective parameter on the “Score Scale, and grade corresponding to the total score position on the “Total Score Scale” was the ultimate prediction [19]. Based on the test results of the model, if the final predicted score is greater than 0.7, it is considered that the content of tea polyphenols is above 35%; when the predicted score is less than 0.4, it is considered that the content of tea polyphenols is below 30%; when the predicted score is between 0.4 and 0.7, it is considered that the content of tea polyphenols is between 30% and 35%.

### 3.4. The Evaluation of the Accuracy and Stability of the Model

Initially used in wars to analyze radar signals, the ROC curve, as an evaluation of the generalization performance of the model, was applied in this study to assess the performance of the model generated by machine learning. The ROC curve separated the results into positive and negative ones. In the final assessment of model accuracy, the actual values were divided into true positive (TP), false negative (FN), false positive (FP), and true negative (TN). TP means the true and estimated values were above 35%, while TN means they were under 30%. FN means the true value was over 35% with the actual value being under 35%, while FP means the true value was under 30% with the actual value being above 30%. Due to the limitation of the ROC curve, the data of the true content of tea polyphenol between 30% and 35% were not adopted in the preliminary accuracy evaluation. However, the previous data were confirmed to have a high prediction accuracy of 83.3% in the ultimate internal test of the model. The abscissa of the ROC curve denoted the true positive chance (TPR), with a greater the value indicating a more accurate prediction.
(4)$$TPR=\frac{TP}{TP+FN’}$$
(5)$$FPR=\frac{FP}{FP+TN’}$$

The closer the ROC curve is to the upper left corner, the bigger the AUC is, which shows a better performance of the prediction model, and the AUC value is generally between 0.5 and 1. In this study, the AUC (area under curve) values of the training set and the validation set were 0.839 and 0.750 respectively, and relevant ROC curves are shown in Figure 4A,B. With AUC > 0.75, the model had better prediction performance, and the result was acceptable. When the AUC value is between 0.5 and 0.75, it is generally considered to be accurate in prediction and has some practical value [20]. The predictive model of this study has been tested and found to have good accuracy, with AUC values greater than 0.750 in both the training and validation sets.

The calibration curve in essence is a scatter plot that could be used to rate the consistency between the predictive and the actual tea polyphenol content. The calibration curve is the visualization of the Hosmer–Lemeshow goodness-of-fit test, comparing the predictive content with the actual content to determine if significant differences are present. In the process of drawing the calibration curve, it is necessary to bin the prediction probability under the guidance of the uniform and quantile. The uniform refers to the same width of each binning, and the quantile refers to the equal number of data points in each binning. Then, the average value of the prediction of all samples in each binning and the probability of a true positive in each binning can be respectively calculated as the abscissa and the ordinate. The calibration curve is created after connecting the points in the scatter plot.

The long dashed line (Ideal) of the calibration curve expresses the ideal one of the nomogram model, and the predicted one is identical to the actual one. Logistic calibration represents the bootstrap-corrected performance of the column chart model, while Nonparametric represents the apparent accuracy of the column chart model [21]. If the prediction value is equal to the true result, the logistic calibration and the nonparametric curve are completely coincident; if the estimated figure is higher than the true one, the nonparametric curve is below the logistic calibration; otherwise, the nonparametric curve is above the logistic calibration. The calibration curve results of the line graph show that the Ideal, Logistic calibration, and Nonparametric lines are closely aligned, indicating a significant consistency in the performance of the tested model in the training and validation sets [22,23] (Figure 5).

### 3.5. System Construction and Model Test

The readability of the nomogram model can be enhanced to some extent by transforming the regression function into the visualized graph, whereas in practice, the nomogram model lacks the ability to rapidly calculate tea polyphenol content. Therefore, this study developed a visualized system based on the nomogram model. It included four modules, namely graphical summary, numerical summary, model summary, and information input. The information input module mainly received the values of three factors, altitude, organic matter, and P, and the prediction could be activated by pressing the button “Predict”. The graphical summary module was responsible for displaying the final prediction value in the form of coordinates. When the mouse moved to the point, six predicted values, altitude, organic matter, P, prediction, lower bound, and upper bound, would be displayed in the chart which could show 11 results simultaneously, and if more prediction results appeared, the old results would be replaced. The numerical summary module would give details about all predictions, including the data displaced in the graphical summary module. The model summary module covering all the parameters in the work was the core part.

Users could directly input the values of altitude, organic matter, and P into the information input module, pressing the “Predict” button to acquire a prediction with lower bound and upper bound values. Similar to the original nomogram model, if the prediction value reaches 0.7 and above, the content of tea polyphenol is considered to be over 35%; if the figure is lower than 0.4, it is less than 30%; otherwise, it is between 30% and 35% (Figure 6).

Data on altitude, organic matter, P and corresponding tea polyphenol content collected from the Yuecheng tea plantation were used to verify the model, and 21 sets of data were applied. Finally, the accuracy rate was 80.95% with the results of 17 sets being correct and the others being wrong, which indicated the high precision of the prediction of the model (Figure 7 and Table 3).

## 4. Conclusions

The study determined and analyzed tea polyphenol contents of teas from various altitudes, soil compositions, and other different conditions in Yuecheng base. The prediction system for tea polyphenol content was constructed with the assistance of the three factors, altitude, organic matter, and P as selected with the LASSO regression model [24]. The evaluation of stability indicated that the AUC values of the training and validation sets were 0.839 and 0.750, respectively, this demonstrates the reliability of the predictive model. Compared with Na Luo et al.’s use of multispectral data to predict the tea polyphenol content of fresh leaves [25], this study could predict the tea polyphenol content of fresh leaves produced based on soil environment before planting. Currently, most research can only predict tea polyphenol content based on fresh leaf yield, meaning ours is a pioneer study in tea tree cultivation. This method of predicting tea polyphenols in pu-erh tea using the soil’s physical and chemical components and altitude is highly beneficial for the selection of tea planting sites in Yunnan. By analyzing local soil composition and altitude, tea growers can utilize this method to identify a relatively suitable tea planting site, thus improving the quality of tea to a certain extent. Compared with the construction of the traditional nomogram model, this research developed a prediction system for the content of pu-erh tea polyphenol, aiming to enhance the ability of quick prediction of the nomogram model. Under the background of an ever-changing global climate, this study explored the change of tea polyphenol content under abiotic stress, laying a solid foundation for further predictions for and studies on the quality of pu-erh tea and providing some theoretical scientific basis.

## Figures and Tables

**Figure 1 foods-12-02128-f001:**
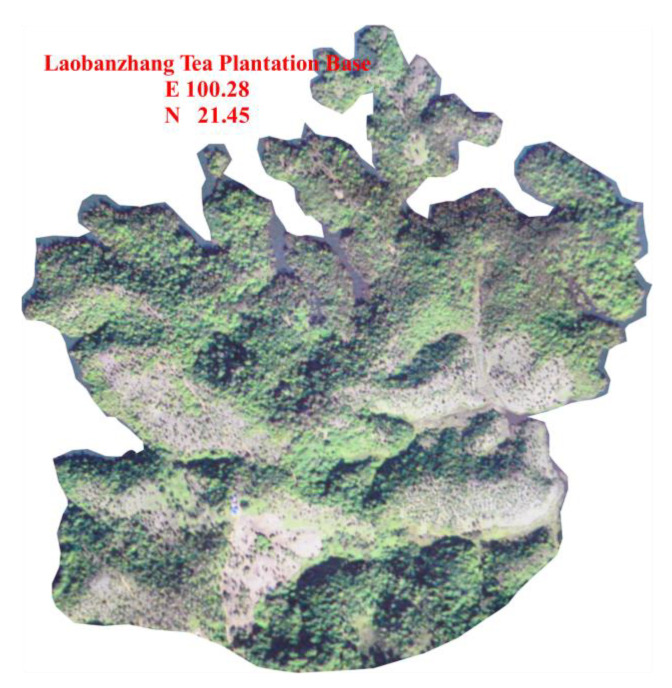
Topographic map of Laobanzhang base.

**Figure 2 foods-12-02128-f002:**
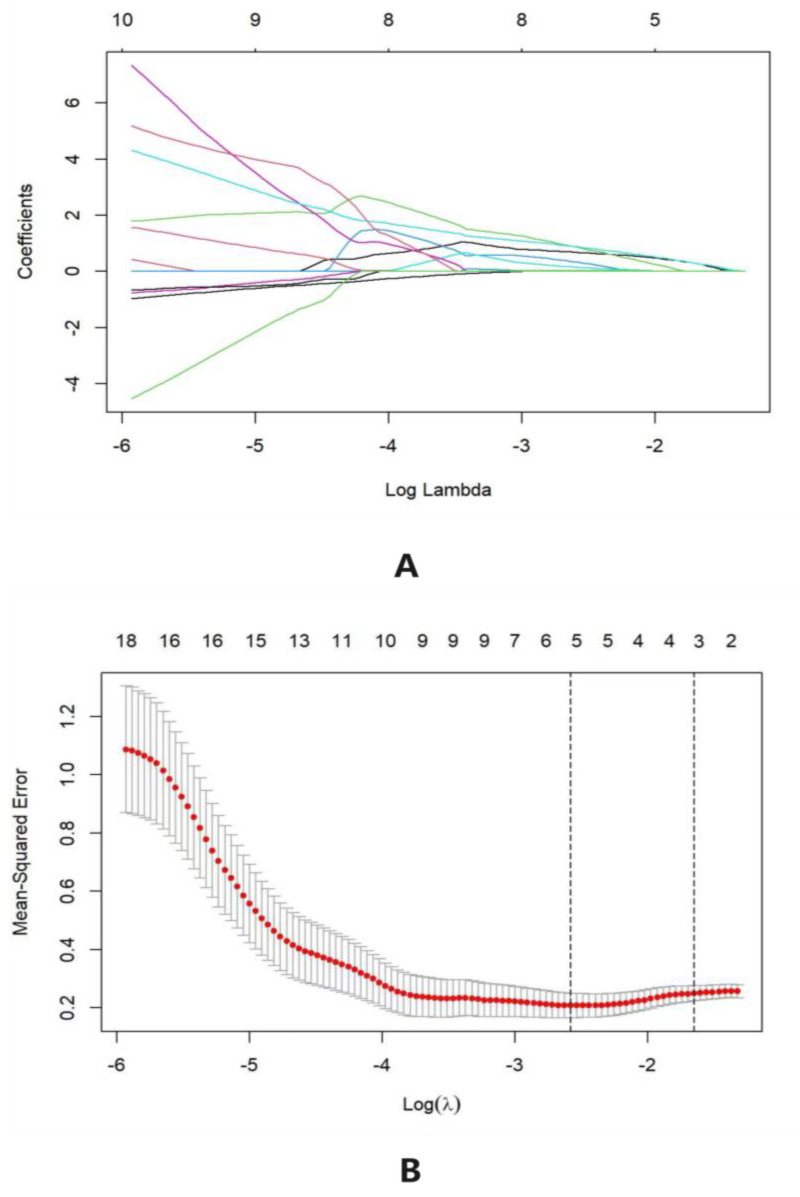
(**A**) The coefficient distribution in LASSO regression, the lines of different colors represent different influencing factors. (**B**) The cross-validation.

**Figure 3 foods-12-02128-f003:**
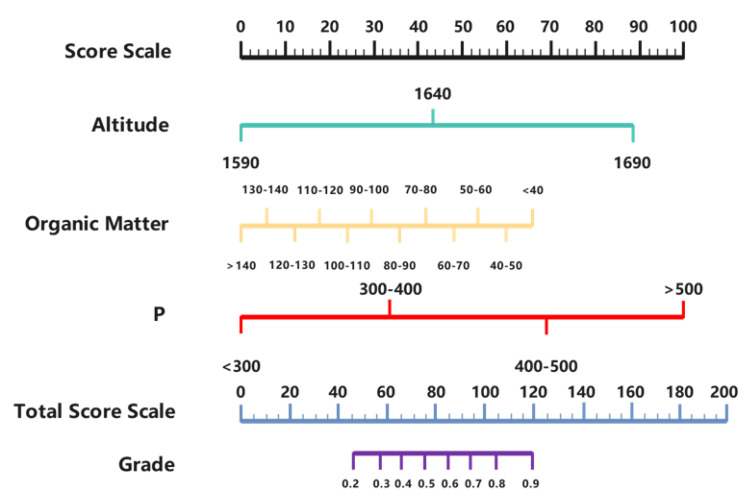
Nomogram for predicting tea polyphenol content.

**Figure 4 foods-12-02128-f004:**
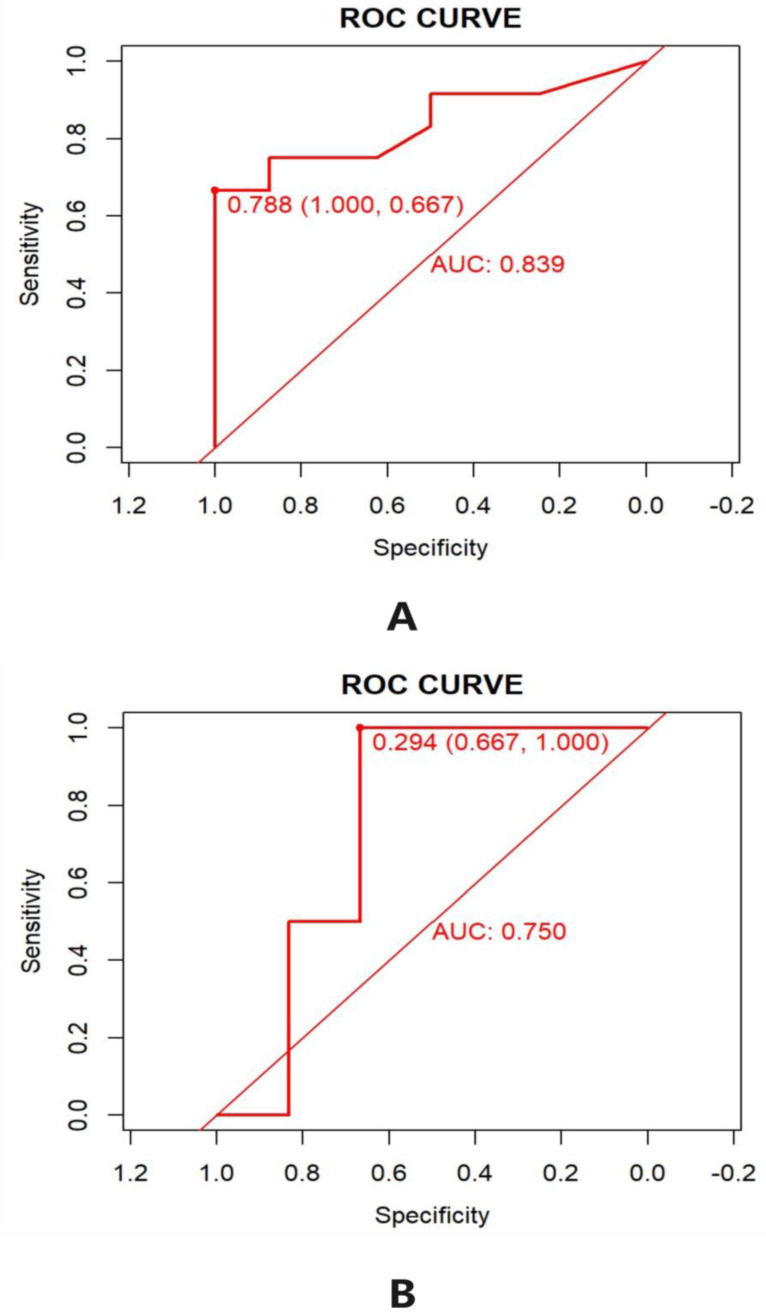
(**A**) The ROC curve analysis of the training set. (**B**) The ROC curve analysis of the test set.

**Figure 5 foods-12-02128-f005:**
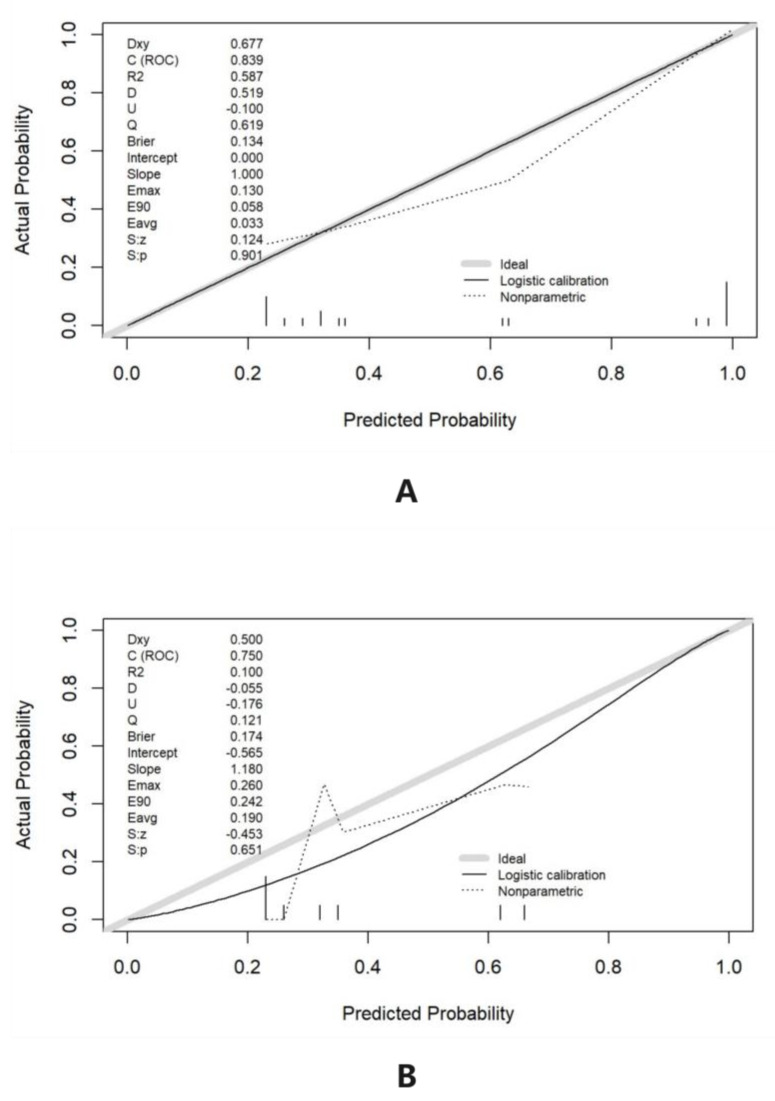
(**A**) The calibration curve analysis of the training set. (**B**) The calibration curve analysis of the test set.

**Figure 6 foods-12-02128-f006:**
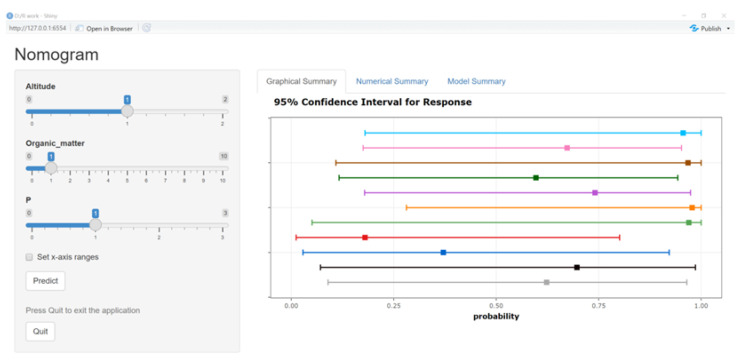
Main interface of the system, The results of each round are represented by different colored lines.

**Figure 7 foods-12-02128-f007:**
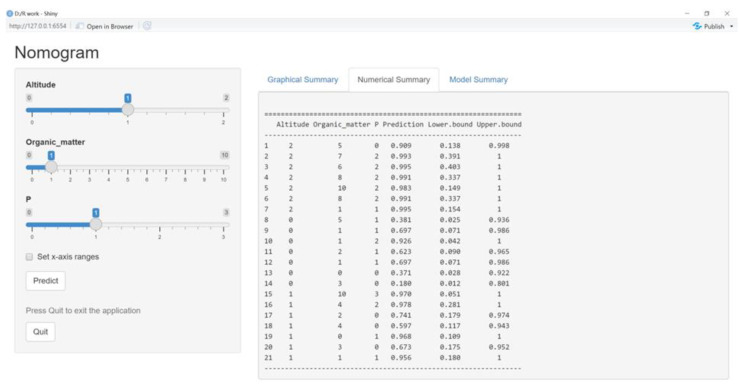
Model test results.

**Table 1 foods-12-02128-t001:** Model size and recognition rate.

Factor Name	B	S.E	Wald	P	Exp(B)	95% CI for EXP(B)	
						Lower	Upper
Altitude	1.673	0.680	6.050	0.014	5.326	1.405	20.191
Arsenic	0.267	0.305	0.769	0.381	1.306	0.719	2.373
Chrome	−0.082	0.235	0.123	0.726	0.921	0.582	1.459
Lead	−0.281	0.169	2.779	0.096	0.755	0.542	1.051
Nickel	1.526	0.746	4.184	0.041	4.600	1.066	19.852
Mercury	1.977	1.178	2.816	0.093	7.222	0.718	72.696
Available Cadmium	1.884	0.898	4.398	0.036	6.579	1.131	38.265
Available Chromium	0.509	0.461	1.215	0.270	1.663	0.673	4.109
Available Nickel	0.056	0.138	0.165	0.684	1.058	0.807	1.386
PH	0.204	0.454	0.202	0.653	1.226	0.503	2.988
Zn	−0.155	0.558	0.077	0.781	0.857	0.287	2.555
Cu	0.123	0.141	0.753	0.386	1.131	0.857	1.492
Organic Matter	0.482	0.203	5.604	0.018	1.619	1.086	2.411
N	1.146	0.483	5.638	0.018	3.147	1.222	8.107
P	1.452	0.632	5.279	0.022	4.272	1.238	14.744
K	−0.523	0.249	4.417	0.036	0.593	0.364	0.965
Available Potassium	1.248	0.719	3.016	0.082	3.484	0.852	14.250
Available Phosphorus	0.525	0.534	0.970	0.325	1.691	0.594	4.813
Alkaline Hydrolysis Nitrogen	1.466	0.708	4.293	0.038	4.332	1.082	17.335
Mg	−0.507	0.399	1.613	0.204	0.602	0.275	1.317
Fluoride	−0.134	0.519	0.067	0.796	0.875	0.316	2.419
Cation	0.111	0.471	0.055	0.814	1.117	0.443	2.814

In the table, B stands for coefficient, S.E for standard error, Wald for Chi-square value, and Exp(B) for OR value.

**Table 2 foods-12-02128-t002:** Multivariate analysis results of factors selected by LASSO regression.

Factor Name	B	S.E	Wald	P	Exp(B)	95% CI for EXP(B)	
						Lower	Upper
Organic Matter	0.155	0.373	1.17	0.677	1.17	0.56	2.43
P	1.722	1.126	5.60	0.126	5.60	0.62	50.86
Altitude	1.888	1.428	6.61	0.186	6.61	0.4	108.52

**Table 3 foods-12-02128-t003:** Model test results.

Altitude (m)	Organic Matter (g/kg)	P (mg/kg)	Tea Polyphenol (mg/kg)	Grade	Correct
1690	88.9	300	39.53	0.909	√
1690	104	499	38.87	0.993	√
1690	97.4	411	40.06	0.995	√
1690	113	490	39.42	0.991	√
1690	163	429	38.28	0.983	√
1690	112	499	38.4	0.991	√
1690	41.4	359	42.99	0.995	√
1590	82.6	315	27.85	0.381	√
1590	43.8	353	34.91	0.697	√
1590	42.7	417	38.14	0.926	√
1590	51.6	336	32.08	0.623	√
1590	42.5	394	30.42	0.697	√
1590	21	42.4	24.53	0.371	√
1590	65.5	113	40.54	0.18	
1640	144	587	38.15	0.97	√
1640	76.2	420	35.09	0.978	√
1640	55.7	214	37.42	0.741	√
1640	74.3	34.8	25.99	0.597	
1640	23.5	367	32.77	0.968	
1640	64.6	274	34.17	0.673	√
1640	49.6	366	34.18	0.956	√

## Data Availability

Publicly available datasets were analyzed in this study. This data can be found here: [https://github.com/anqi99/Nomogram-Model.git (accessed on 14 January 2023)].

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
