# Peer review of "A Nomogram Model for Predicting the Polyphenol Content of Pu-Erh Tea"

_foods, 2023, doi:10.3390/foods12112128_

Round 1
Reviewer 1 Report
The article is about the prediction studying for the content of Pu-erh tea polyphenol based on the Nomogram model. It also focused on the effect of abiotic stress and contents of Pu-erh tea polyphenol in teas produced by Yuecheng, a Xishuang banna-based tea producer in Yunnan province. It can be considered as applied research and I think the subject is overall interesting. The manuscript is well structured. The article needs these changes.
Abstract: Abstract can be increased adding one or two lines of methodology and future impact of the study.
Line 37: Add and remove extra space; Soil is the growth substrate for tea trees Camellia sinensis( L.)O. Kuntze.]
Line 37-40: Rewrite the sentences for clarity. Normally, Sentence does not start with “and”
Soil is the growth substrate for tea trees Camellia sinensis ( L.)O. Kuntze.] so that
soil fertility, as the comprehensive reflection of soil indicators, determines the growth of
tea trees. And soil nutrients, vital to soil fertility, weigh heavily in the quality and pro duction of tea.
Line 49: Correction: Many factors influence the tea polyphenol content of fresh tea leaves, such as temperature, light, water, air and so on, effecting growth and economic production of tea trees, and the formation, metabolism and transformation of biochemical components in tea.
Line 45, 47, 57,59, 64, 66, 72, 74 and so on: Add space before every citation.
Line 107, 126, 193: Add space before paratheses he Yuecheng base (Figure 1)
Line 108, 100: PL. Add the references at the end of sentence; Fresh leaves were picked under the one-bud-with-two-leaves standard and processed under the guidance of the conventional process of sun dried mao cha.
Line 116-140: All the parameters along with detailed procedures should be discussed separately; the content of matters soaked out in water constant weight method, the gross
amount of amino acids ninhydrin colorimetry, tea polyphenol content Folin phenol col-
orimetry, and catechin and caffeine contents High Performance Liquid Chromatography
(HPLC) established in the lab[11]. The determination of copper, zinc, chromium, nickel
and lead was carried out by Flame atomic absorption spectrophotometry, the determination of total phosphorus by alkali fusion-Mo-Sb Anti spectrophotometric method,
Line 142, 143: Pl. Add details of province and country; AMD Ryzen R7-5800 H processor, 16 GBRAM, 1TB hard disk and 16 GB NAIDIA RTX 3060 GPU, NVIDIA 461.37
driver, CUDA 11.0 version, R language is 4.1.2 version CUDA 11.0 version, R language is 4.1.2 version.
What is the control used during all the research work. Pl. elaborates.
Line 218: Use either By or using; Both are used for same. By or using random sampling
with replacement to get enough samples from original data sets
Line 223, 224: Add details of the software provider; The multivariate analysis results of Logistic regression by the software Rstudio are listed in Table 2.
Line 266, 267, 289, 290: Illustrate Figure 4- & 5-part A and B in the caption.
Conclusion: Conclusion should be short and comprehensive
Author Response
Thanks very much for your time to review this manuscript. I really appreciateyou’re your comments and suggestions. We have considered these comments carefully and triedour best to address every one of them.
- Abstract: Abstract can be increased adding one or two lines of methodology and future impact of the study.
Modification instructions: Have been changed in accordance with the advice given. This research explored the change of tea polyphenol content under abiotic stress, laying a solid foundation for further predictions for and studies on the quality of Pu-erh tea and providing some theoretical scientific basis.
- Line 37: Add and remove extra space; Soil is the growth substrate for tea trees Camellia sinensis( L.)O. Kuntze.]
Modification instructions: Have been changed in accordance with the advice given.
- Line 37-40: Rewrite the sentences for clarity. Normally, Sentence does not start with “and” Soil is the growth substrate for tea trees Camellia sinensis ( L.)O. Kuntze.] so that soil fertility, as the comprehensive reflection of soil indicators, determines the growth of tea trees. And soil nutrients, vital to soil fertility, weigh heavily in the quality and production of tea.
Modification instructions: Have been changed in accordance with the advice given.
- Line 49: Correction: Many factors influence the tea polyphenol content of fresh tea leaves, such as temperature, light, water, air and so on, effecting growth and economic production of tea trees, and the formation, metabolism and transformation of biochemical components in tea.
Modification instructions: Have been changed in accordance with the advice given.
- Line 45, 47, 57,59, 64, 66, 72, 74 and so on: Add space before every citation. Line 107, 126, 193: Add space before paratheses he Yuecheng base (Figure 1)
Modification instructions: Have been changed in accordance with the advice given.
- Line 108, 100: PL. Add the references at the end of sentence; Fresh leaves were picked under the one-bud-with-two-leaves standard and processed under the guidance of the conventional process of sun dried mao cha.
Modification instructions: Have been changed in accordance with the advice given.
- Line 116-140: All the parameters along with detailed procedures should be discussed separately; the content of matters soaked out in water constant weight method, the gross amount of amino acids ninhydrin colorimetry, tea polyphenol content Folin phenol colorimetry, and catechin and caffeine contents High Performance Liquid Chromatography (HPLC) established in the lab[11]. The determination of copper, zinc, chromium, nickel and lead was carried out by Flame atomic absorption spectrophotometry, the determination of total phosphorus by alkali fusion-Mo-Sb Anti spectrophotometric method,
Modification instructions: Have been changed in accordance with the advice given.
- Line 142, 143: Pl. Add details of province and country; AMD Ryzen R7-5800 H processor, 16 GBRAM, 1TB hard disk and 16 GB NAIDIA RTX 3060 GPU, NVIDIA 461.37 driver, CUDA 11.0 version, R language is 4.1.2 version CUDA 11.0 version, R language is 4.1.2 version. What is the control used during all the research work. Pl. elaborates.
Modification instructions: Have been changed in accordance with the advice given. The model was built on a computer produced by Lenovo (Beijing) Co., Ltd in China, with an AMD Ryzen R7-5800H processor, 16GB RAM, 1TB solid-state drive, and NVIDIA GeForce RTX 3060 Laptop GPU. The operating system used is Windows 10, with CUDA 11.0 version, and R language version 4.1.2.
- Line 218: Use either By or using; Both are used for same. By or using random sampling with replacement to get enough samples from original data sets
Modification instructions: Have been changed in accordance with the advice given.
- Line 223, 224: Add details of the software provider; The multivariate analysis results of Logistic regression by the software Rstudio are listed in Table 2.
Modification instructions: Have been changed in accordance with the advice given.
- Line 266, 267, 289, 290: Illustrate Figure 4- & 5-part A and B in the caption.
Modification instructions: Have been changed in accordance with the advice given. Figure 4A shows the ROC curve analysis of the training set, Figure 4B shows the ROC curve analysis of the test set. Figure 5A shows the calibration curve analysis of the training set, Figure 5B shows the calibration curve analysis of the test set.
- Conclusion: Conclusion should be short and comprehensive
Modification instructions: Have been changed in accordance with the advice given.

Reviewer 2 Report
The manuscript proposes a novel procedure for predicting Pu-erh tea polyphenol content using the nomogram model. The manuscript is generally well-written.
The objective and novelty should be indicated in more detail in the Abstract.
Species names in Latin should be italicized.
line 83: The authors mentioned artificial intelligence. This issue should be expanded.
line 107: Why the variety Mengku and only one variety was chosen?
The number of repetitions should be given for each type of measurement.
The obtained results should be compared with other literature data. The discussion should be expanded.
Author Response
Thanks very much for your time to review this manuscript. I really appreciateyou’re your comments and suggestions. We have considered these comments carefully and triedour best to address every one of them.
- The objective and novelty should be indicated in more detail in the Abstract.
Modification instructions: Have been changed in accordance with the advice given. This research explored the change of tea polyphenol content under abiotic stress, laying a solid foundation for further predictions for and studies on the quality of Pu-erh tea and providing some theoretical scientific basis.
- Species names in Latin should be italicized.
Modification instructions: Have been changed in accordance with the advice given.
- line 83: The authors mentioned artificial intelligence. This issue should be expanded.
Modification instructions: Have been changed in accordance with the advice given.
- line 107: Why the variety Mengku and only one variety was chosen?
Modification instructions: The reason for selecting Mengku large-leaf tea tree as the research object is that it is the main variety used for producing Pu-erh tea and has a wide planting area in Yunnan, which makes it representative. The method used in this study has great scalability and can be applied to other varieties in the future by simply modifying the training dataset to achieve more accurate predictions for tea trees in related regions and varieties. Based on our research, there are also differences in tea trees of the same variety in different regions. Therefore, our research method is more suitable for adapting to local conditions and varieties.
- The number of repetitions should be given for each type of measurement.
Modification instructions: Each soil and tea sample was tested for composition three times, and the parameters were determined using the average and standard error of the three tests. Have been changed in accordance with the advice given.
- The obtained results should be compared with other literature data. The discussion should be expanded.
Modification instructions: Compared with Na Luo et al.’s use of multispectral data to predict the tea polyphenol content of fresh leaves, this study can predict the tea polyphenol content of fresh leaves produced based on soil environment before planting. Currently, most research can only predict tea polyphenol content based on fresh leaf yield, making this study pioneering in tea tree cultivation. Have been changed in accordance with the advice given.

Round 2
Reviewer 2 Report
The manuscript has been improved.